# Robust Contrastive Multi-view Clustering against Dual Noisy Correspondence

**Ruiming Guo**[1]*, **Mouxing Yang**[1]*, **Yijie Lin**[1], **Xi Peng**[1,2], **Peng Hu**[1]†
[1]College of Computer Science, Sichuan University, China
[2]State Key Laboratory of Hydraulics and
Mountain River Engineering, Sichuan University, China
{guoruiming.gm, yangmouxing, linyijie.gm, pengx.gm, penghu.ml}@gmail.com

## Abstract

Recently, contrastive multi-view clustering (MvC) has emerged as a promising avenue for analyzing data from heterogeneous sources, typically leveraging the off-the-shelf instances as positives and randomly sampled ones as negatives. In practice, however, this paradigm would unavoidably suffer from the Dual Noisy Correspondence (DNC) problem, where noise compromises the constructions of both positive and negative pairs. Specifically, the complexity of data collection and transmission might mistake some unassociated pairs as positive (namely, false positive correspondence), while the intrinsic one-to-many contrast nature of contrastive MvC would sample some intra-cluster samples as negative (namely, false negative correspondence). To handle this daunting problem, we propose a novel method, dubbed Contextually-spectral based correspondence refinery (CANDY). CANDY dexterously exploits inter-view similarities as *context* to uncover false negatives. Furthermore, it employs a spectral-based module to denoise correspondence, alleviating the negative influence of false positives. Extensive experiments on five widely-used multi-view benchmarks, in comparison with eight competitive multi-view clustering methods, verify the effectiveness of our method in addressing the DNC problem. The code is available at https://github.com/XLearning-SCU/2024-NeurIPS-CANDY.

## 1 Introduction

In real-world applications, data are often presented in various modalities or views, including but not limited to visible images, thermal images, text, and audio [1, 2]. Multi-view Clustering (MvC), a fundamental tool in multi-view data analysis, aimed at learning a common space in which data are grouped into distinct clusters, attracts significant attention across various research communities [3–9]. In recent years, contrastive MvC methods have emerged as a central focus in multi-view clustering researches [10, 11]. The typical implementation of these methods involves leveraging the off-the-shelf data pairs as positives and randomly sampling cross-view pairs as negatives, followed by employing contrastive learning upon them [12–14]. As a result, the cross-view discrepancy could be eliminated, revealing the underlying cluster structure.

Although existing contrastive MvC methods have achieved promising performance, their success heavily relies on the assumption of faultless cross-view correspondence. In practice, however, this assumption is hard or even impossible to meet [15–19], leading to inevitable contamination of the cross-view correspondence, as shown in Fig. 1a. More specifically, the complexity of data collection

---

*Equal contribution
†Corresponding author

38th Conference on Neural Information Processing Systems (NeurIPS 2024).

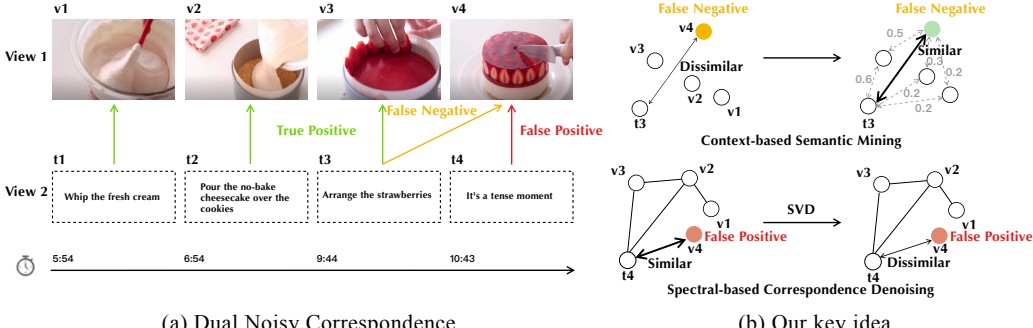

(a) Dual Noisy Correspondence        (b) Our key idea

Figure 1: The motivation and key idea. (a) Dual noisy correspondence. The cross-view data pairs are contaminated by both false positive and negative correspondences, and the clean and noisy correspondence is mixed. (b) Top: Context-based Semantic Mining. The existing studies estimate the data affinity based on the data representation and might neglect the out-of-neighborhood yet semantically-associated false negatives. In contrast, we formulate the affinity from one data point to all the others as the context and use them for similarity induction, thus benefiting the false negative uncovering in a global manner; Bottom: Spectral-based Correspondence Denoising. Borrowing from spectral decomposition for signal denoising, we employ spectral denoising on the contextual affinity graph to prevent false positives from dominating the model optimization. In the figure, the thickness of the black arrows represents the association strength between two data points.

and transmission might mistake certain unaligned cross-view pairs for positive pairs, leading to false positive correspondence. Conversely, the inherent one-to-many contrast characteristic of contrastive MvC would inevitably result in semantically-associated cross-view positives being wrongly treated as negatives, thus producing false negative correspondence.

Based on the above observations, this paper reveals a novel problem for contrastive MvC called Dual Noisy Correspondence (DNC). Formally, DNC refers to the noise present in both cross-view positive and negative pairs. This problem is akin to the *partially view-aligned problem* (PVP), yet differs in that PVP presupposes the availability of some correctly-associated instances for training, while DNC breaks through this impractical assumption and remains agnostic to any clean correspondence [20, 21]. Thus, DNC could be regarded as a more practical yet challenging variant of PVP, resulting in the infeasibility of PVP-oriented methods to address the DNC problem. Notably, our experimental findings, detailed in Section 4, support this claim.

To tackle the DNC problem, we present a novel robust method, dubbed ContextuAlly-spectral based correspoNDence refinerY (CANDY), for learning to cluster with noisy positive and negative correspondences. As illustrated in Fig. 1b, CANDY consists of two core modules: i) the Context-based Semantic Mining (CSM) module for recalling the false negatives, and ii) the Spectral-based Correspondence Denoising (SCD) module for alleviating the adverse impact of false positives. To be specific, CANDY first constructs a cross-view affinity graph from the multi-view data. Subsequently, CANDY calculates the connection probabilities from each node to all others, forming the context, and exploits CSM to induce a high-order contextual affinity graph. Thanks to the properties of high-order affinity, CSM could facilitate the discovery of semantically-associated positives hidden in the negatives. After that, inspired by singular value decomposition techniques used in image denoising [22, 23], CANDY performs spectral decomposition on the contextual affinity graph and employs SCD to filter noise in the graphical spectrum, thus mitigating overfitting to false positives. Finally, CANDY employs the denoised contextual affinities to weight arbitrary contrastive losses to achieve robust MvC against DNC.

In summary, the main contributions and novelties of this work could be summarized as follows.

- We reveal and study a new practical problem in contrastive multi-view clustering, namely, dual noisy correspondence (DNC). Unlike prior PVP-oriented studies that rely on quite a few correctly-associated pairs, DNC refers to noise inherent in both cross-view positive and negative pairs. To the best of our knowledge, this could be one of the first investigations into noisy correspondence within MvC, particularly the more practical and challenging DNC problem.

- We propose a novel robust method called CANDY for enhancing the robustness of contrastive MvC against DNC, embracing the following novelties: i) The formulation of affinity from one data point to others as context, facilitating the revelation of false negatives; and ii) Spectral denoising upon the high-order affinity graph, preventing overfitting to false positives.

- Extensive experiments verify the effectiveness and superiority of CANDY. Moreover, we demonstrate the generalizability of CANDY, showing that it could serve as a plug-and-play solution to enhance the robustness of most contrastive MvC methods against DNC.

## 2 Related Work

In this section, we present a brief review of two topics related to this work: multi-view clustering and noisy correspondence learning.

### 2.1 Contrastive Multi-view Clustering

The inherent pairing characteristic of the multi-view data renders the contrastive learning paradigm a natural fit for MvC, giving rise to the established paradigm of contrastive MvC. Existing MvC methods could be roughly classified into the following three groups: i) Vanilla contrastive MvC methods [24], which directly exploit contrastive learning to enhance the discrimination of learned representations by maximizing the mutual information between distinct views. ii) Robust contrastive MvC methods against incomplete instances [25–27], which employ contrastive learning to learn the cross-view consistency, thereby facilitating the recovery of missing samples. iii) Robust contrastive MvC methods against false negatives [18, 28], which redesign dedicated loss functions or similarity estimation techniques to conquer false negatives inherent in contrastive learning, thus boosting clustering performance.

Our CANDY, alongside the works of [18, 28], is devoted to addressing false negatives, while having the following significant distinctions. Different from [18], which utilizes a false-negative-robust loss, CANDY presents a Context-based Semantic Mining (CSM) module to induce a context-aware and high-order affinity graph, benefiting the discovery of false negatives from a global perspective. Moreover, [28] proposes modeling the probability of false negatives by resorting to random walks while being susceptible to cross-view false positives. In contrast, thanks to the SCM module, CANDY embraces a more robust performance in uncovering false negatives, as verified in our experimental results.

### 2.2 Noisy Correspondence Learning

In the era of big data, millions of multimodal data are crawled from the Internet, often requiring extensive curation, which is time-intensive and cost-prohibitive [15–17]. Nevertheless, it is almost impossible to eliminate misalignment in a large quantity of multimodal data, leading to noisy correspondence. To handle this problem, noisy correspondence learning is presented to alleviate the negative influence of false positive and negative correspondences within data pairs, which has achieved promising results across various applications, such as cross-modal retrieval [29–31], object re-identification [32–34], multi-view learning [21, 35], graph matching [36], video reasoning [37], image-text pre-training [38].

To the best of our knowledge, this work could be one of the first studies on learning to cluster with noisy correspondence. Unlike most existing approaches focusing solely on either false positives or negatives [30, 21], our CANDY addresses the more general challenge called Dual Noisy Correspondence (DNC). Extensive experiments reveal the impracticality of applying the existing approaches to DNC in MvC, highlighting the necessity of a tailored solution to MvC against the DNC problem.

## 3 Method

In this section, we elaborate on the proposed ContextuAlly-spectral based correspoNDence refinerY (CANDY), which aims to enhance the robustness of contrastive MvC against the Dual Noisy Correspondence (DNC) problem. As illustrated in Fig. 2, our CANDY consists of two novel modules:

Figure 2: Overview of CANDY. First, each view is fed into a view-specific encoder to generate the embeddings. These embeddings are adopted to construct both inter- and intra-view affinity graphs, with edges weighted by Gaussian kernel similarity. The context-based semantic mining module dexterously reformulates inter-view similarities as "context", employing this context as a set of bases to induce a new contextual affinity space. In this space, the rooted/dissimilar false negatives could be brought to light. Second, the spectral-based correspondence denoising module steps in to alleviate the adverse impacts of noisy correspondence on positive pairs, thus obtaining a low-noise pseudo target. Finally, this pseudo target steers the contrastive learning process, enhancing robustness against DNC in MvC. For the sake of brevity, this figure only presents a simplified depiction involving two views, and the robust contrastive MvC from view 1 to view 2.

a context-based semantic mining module to uncover inherent false negatives, and a spectral-based correspondence denoising module to prevent contrastive MvC from overfitting false positives. In the following, we commence with the mathematical formulation of the DNC problem in Section 3.1, proceed to the context-based semantic mining module in Section 3.2, and culminate with the spectral-based correspondence denoising module in Section 3.3.

## 3.1 Problem Formulation

Given the multi-view dataset $D = \{(\mathbf{x}_i^{(1)}, \ldots, \mathbf{x}_i^{(V)})\}_{i=1}^N$ with $N$ instances observed from $V$ views, the objective of contrastive MvC is to group these instances into $K$ clusters. To this end, contrastive MvC methods construct the sets of positive and negative pairs as $\bigcup_{i=1}^N \left\{ \left( \mathbf{x}_i^{(v_1)}, \mathbf{x}_i^{(v_2)}, c_i \right) \mid c_i = 1, 1 \leq v_1, v_2 \leq V, v_1 \neq v_2 \right\}$ and $\bigcup_{\substack{i=1 \\ i \neq j}}^N \bigcup_{j=1}^N \left\{ \left( \mathbf{x}_i^{(v_1)}, \mathbf{x}_j^{(v_2)}, c_i \right) \mid c_i = 0, 1 \leq v_1, v_2 \leq V, v_1 \neq v_2 \right\}$ by utilizing the off-the-shelf instances and perform random sampling across views respectively, where $c$ denotes the established cross-view correspondence. Subsequently, the contrastive loss [39, 40] is applied to eliminate the cross-view discrepancy and reveal the cluster structure. However, as elaborated in the Introduction, cross-view correspondence could often be contaminated by both false positives and negatives. More specifically, a certain amount of unassociated ($\hat{c} = 0$) and associated ($\hat{c} = 1$) pairs would be wrongly treated as positives ($c = 1$) and negatives ($c = 0$) respectively, while the ground-truth correspondence $\hat{c}$ is unknown. In particular, the ratio of false negatives would reach up to $1/k$ when the categories of the dataset $D$ are uniformly distributed, where $k$ is the number of classes.

To counter the DNC challenge, we introduce a soft contrastive loss:

$$\mathcal{L} = \sum_{v_1=1}^V \sum_{\substack{v_2=1 \\ v_2 \neq v_1}}^V \mathcal{H} \left( \mathbf{C}^{(v_1, v_2)}, \rho \left( \mathbf{Z}^{(v_1)} \mathbf{Z}^{(v_2)^\top} \right) \right), \tag{1}$$

where $\mathcal{H}$ denotes the row-wise *cross-entropy* function with mean reduction, $\mathbf{C}^{(v_1, v_2)} \in \mathbb{R}^{n \times n}$ is the pseudo target (Eq. 6), $\mathbf{Z}^{(v_1)} \mathbf{Z}^{(v_2)^\top}$ represents the affinity matrix between views $v_1$ and $v_2$, and $\rho(\cdot)$ signifies the *softmax* function. The batch-wise representation matrix $\mathbf{Z}^{(v)} \in \mathbb{R}^{n \times d}$ encapsulates features extracted by the view-specific encoder $f^{(v)}$, with $n$ denoting the batch size. The *softmax*

function ($\rho(\cdot)$) is applied row-wise to ensure each row sums to one as follows:

$$\left[\rho\left(\mathbf{Z}^{(v_1)}\mathbf{Z}^{(v_2)\top}\right)\right]_{ij} = \frac{\exp\left(\left[\mathbf{Z}^{(v_1)}\right]_i \left[\mathbf{Z}^{(v_2)}\right]_j^\top /\tau\right)}{\sum_{t=1}^n \exp\left(\left[\mathbf{Z}^{(v_1)}\right]_i \left[\mathbf{Z}^{(v_2)}\right]_t^\top /\tau\right)}. \tag{2}$$

In general, traditional contrastive MvC methods assume that the cross-view correspondence is faultless, typically adopting an identity matrix $\mathbf{I} \in \mathbb{R}^{n \times n}$ as the target. As verified in our experiments, such a vanilla target not only misleads the model to overfit false positives but also neglects numerous semantically associated false negatives. Therefore, the goal of CANDY becomes generating a robust pseudo target resilient against the DNC problem.

## 3.2 Context-based Semantic Mining

The crux of uncovering false negatives lies in accurately modeling the semantic association between data points. Therefore, the widely-used strategy is based on the point-to-point similarity in the affinity graph. Specifically, a fully-connected affinity graph $\mathbf{A}$ is first constructed using the feature $\mathbf{Z}^{(v_1)}$ and $\mathbf{Z}^{(v_2)}$ as nodes in a mini-batch, with edge weights defined by Gaussian kernel similarity. Mathematically,

$$\mathbf{A}_{ij}^{(v_1 \to v_2)} = \exp\left(-\left\|\left[\mathbf{Z}^{(v_1)}\right]_i - \left[\mathbf{Z}^{(v_2)}\right]_j\right\|^2 /\sigma\right), \tag{3}$$

where $\sigma$ is a scale parameter and $v_1$ is the anchor view. After that, a cross-view graph $\hat{\mathbf{A}}^{(v_1 \to v_2)}$, where each edge represents the probability of semantic association between the corresponding two nodes, could be obtained by normalizing $\mathbf{A}$ in a row-wise manner. This strategy, however, tends to be short-sighted, potentially neglecting the out-of-neighborhood yet semantically-associated false negatives, as shown in Fig. 1b and supported by our experiments.

In contrast, a simple yet effective semantic modeling strategy is presented to formulate the connection probability from one node to all others as a context, thereby redefining the context as a special representation for semantic mining. Intuitively, the context $\hat{\mathbf{A}}_{i\cdot}^{(v_1 \to v_2)} = \left[\hat{\mathbf{A}}_{i1}^{(v_1 \to v_2)}, \cdots, \hat{\mathbf{A}}_{in}^{(v_1 \to v_2)}\right]$ serves as a new embedding for the node $i$, facilitating the construction of a cross-view high-order affinity graph $\mathbf{G}^{(v_1 \to v_2)}$ as follows:

$$\mathbf{G}^{(v_1 \to v_2)} = \mathbf{A}^{(v_1 \to v_2)}\mathbf{A}^{(v_2 \to v_2)\top} \tag{4}$$

Thanks to context modeling, our CSM embraces two distinct advantages: i) it encapsulates the structural information of nodes into the graph, enhancing the ability of global semantic mining, and ii) it provides a novel basis for data representation to project nodes into a new affinity space, potentially better uncovering semantically-associated false negatives.

## 3.3 Spectral-based Correspondence Denoising

The false positive correspondence would emerge in both the off-the-shelf positive pairs, as elaborated in Section 1, and the wrongly associated negatives during the construction of $\mathbf{G}^{(v_1 \to v_2)}$. To address this, we propose a correspondence-denoising mechanism for the high-order affinity graph $\mathbf{G}^{(v_1 \to v_2)}$ based on the spectral denoising theorems [22, 23]. In brief, it is widely acknowledged that the eigenvectors of signals corresponding to larger eigenvalues represent principal components, while smaller ones are apt to be noise. By selectively discarding the information tied to the minor eigenvalues, one could filter out the noise, thereby revealing the underlying structures. Inspired by these preliminary insights, we propose refining $\mathbf{G}^{(v_1 \to v_2)}$ by resorting to *singular value decomposition*. Mathematically,

$$\mathbf{G}^{(v_1 \to v_2)} = \mathbf{U}\mathbf{\Sigma}\mathbf{V}^\top, \tag{5}$$

where $\mathbf{\Sigma}$ denotes a diagonal matrix consisting of the singular values, $\mathbf{U}$ and $\mathbf{V}$ is the left singular matrix and the right singular matrix, respectively.

After that, the denoised pseudo target could be obtained via

$$\widetilde{\mathbf{G}}^{(v_1 \to v_2)} = \mathbf{U}\widetilde{\mathbf{\Sigma}}\mathbf{V}^\top, \tag{6}$$

where $\widetilde{\Sigma} = \mathrm{diag}(\lambda_1, \cdots, \lambda_L)$ is a diagonal matrix consisting of the retained singular values ($\lambda_1 > \cdots > \lambda_L \geq \eta$), with $\eta$ being a denoising hyper-parameter fixed as $0.2$ in our experiments.

By combining the denoised pseudo target with the vanilla target ($\mathbb{I}$), we obtain the noise-resisted pseudo target ($\mathbf{C}^{(v_1, v_2)}$) for the proposed soft contrastive loss (Eq. 1) via

$$\mathbf{C}^{(v_1, v_2)} = \lambda \mathbf{I} + \widetilde{\mathbf{G}}^{(v_1 \rightarrow v_2)}, \tag{7}$$

where $\lambda$ is fixed as $0.2$ in our experiments.

## 4 Experiments

In this section, we verify the effectiveness of our CANDY against the DNC problem through extensive experiments by addressing the following questions:

1. **Performance Superiority**: Does CANDY outperform the existing state-of-the-art (SOTA) MvC methods, including those designed for PVP?

2. **Component Indispensability**: Are all components crucial for maintaining robustness against DNC?

3. **Working Mechanism**: How does CANDY achieve robustness against DNC?

4. **Approach Necessity**: Why is it necessary to design an approach for the DNC problem instead of using existing noisy correspondence learning methods?

5. **Approach Generalizability**: Can CANDY be used in a plug-and-play manner to endow other contrastive MvC methods with robustness against DNC?

### 4.1 Configurations and Implementation Details

CANDY is designed as a plug-and-play solution to endow most existing contrastive MvC methods with robustness against the DNC problem. Therefore, we choose the SOTA contrastive MvC method, namely, DIVIDE [28], as our baseline. Specifically, we retain the architecture and pipeline of DIVIDE, modifying only the loss function. Following DIVIDE, to obtain a good initialization for the neural networks, we use the vanilla contrastive loss by setting the target $\mathbf{C}^{(v_1, v_2)}$ in Eq. 1 as the identity matrix $\mathbf{I}$ for the first 20 epochs of training. To endow DIVIDE with robustness against DNC, we incorporate context-based semantic mining and spectral-based correspondence denoising modules, alongside the soft contrastive loss (Eq. 1). Since MvC requires training and clustering on the same dataset, we conduct the view realignment strategy on the learned representation by following the PVP studies [20, 21]. For achieving clustering, we concatenate the realigned representations across views to form a common representation of the MvC data and then apply the $k$-means algorithm by following [25].

In the experiment, CANDY is implemented with PyTorch 2.1.2, and the model is optimized with the Adam [41] optimizer with a learning rate of $0.002$ across all experiments, with a batch size fixed to 1024. All evaluations are conducted on Ubuntu 20.04 OS with NVIDIA 3090 GPUs. The scale parameter $\sigma$ in Eq. 3 is fixed as $0.07$ across all experiments. The experiments are carried out on the following five widely-used multi-view learning datasets.

- **Scene-15** [42] includes 4,485 images across 15 categories. We employ PHOG and GIST as two distinct views following [18].

- **Caltech-101** [43] consists 8,677 images collected from 101 classes. We use two kinds of deep features extracted by the DECAF and VGG19 neural networks as two views following [44].

- **LandUse-21** [45] contains 2,100 satellite imagery samples in 21 categories. We employ the PHOG and LBP features as two views following Lin et al. [46].

- **Reuters** [47] is a repository of news content in multiple languages with 18,758 samples. Following [48], we transform the texts into a 10-dimensional latent space with a conventional autoencoder and use English and French as two different views.

Table 1: The statistics of false positive and false negative ratios (%) with respective to different datasets and $\eta$ in the experiments.

| $\eta$ | Caltech101 | | LandUse21 | | NUSWIDE | | Reuters | | Scene15 | |
|---|---|---|---|---|---|---|---|---|---|---|
| | FP | FN | FP | FN | FP | FN | FP | FN | FP | FN |
| 0.0 | 0.00 | 2.84 | 0.00 | 4.73 | 0.00 | 9.99 | 0.00 | 21.40 | 0.00 | 6.91 |
| 0.2 | 19.34 | 2.84 | 19.10 | 4.73 | 17.98 | 9.99 | 15.67 | 21.40 | 18.68 | 6.91 |
| 0.5 | 48.45 | 2.84 | 47.33 | 4.73 | 45.07 | 9.99 | 39.64 | 21.40 | 46.42 | 6.91 |
| 0.8 | 77.48 | 2.84 | 76.24 | 4.73 | 72.03 | 9.99 | 62.57 | 21.40 | 73.98 | 6.91 |

- **NUS-WIDE** [49] includes 9,000 images paired with their respective captions from 10 classes. We adopt a VGG19 neural network for the extraction of visual features, and a Sentence CNN to extract the text features by following [50].

For comprehensive evaluations, we vary the noise ratio in the datasets by adopting the following protocols. For the false positive correspondence, we select one view as the anchor and randomly shuffle samples in other views according to the specified FP ratio $\eta$ which is varied from $0\%$, $20\%$, $50\%$, to $80\%$. For the false negative correspondence, we adhere to the inherent FN ratio in each dataset. For clarity, we present the statistics of FP and FN ratios for different datasets in Table 1. Notably, as the samples within the same instance would be regarded as negative if they do not belong to the same class, the practical FP ratios might be slightly lower than the specified $\eta$.

### 4.2 Comparison with State of the Arts (Performance Superiority)

In this section, we compare CANDY with eight SOTA MvC methods including the typical MvC methods (DCCAE [51], BMVC [52]), the PVP-oriented MvC methods (MvCLN [21], PVC [20], SURE [18], CGCN [53]), the false-negative-robust contrastive MvC (GCFAgg [54], and DI-VIDE [28]). Following the widely-used evaluation protocols, we adopt "ACC", "NMI" and "ARI" as the metrics.

Table 2 presents the comparison results for each dataset and the average results overall, where one could have the following observations. First, our CANDY outperforms all baselines in terms of the average ACC and ARI when the FP ratio is $0\%$, which could be attributed to the powerful semantic mining capacity on the false negatives. Second, all baselines experience heavy performance degradation when encountering false positives. In contrast, CANDY achieves significant robustness and remarkably outperforms all baselines by a large margin. The above two observations could verify the effectiveness of CANDY against the DNC problem.

Furthermore, we explore the capacity of CANDY on handling the other important problem in MvC, namely, missing views. To this end, we follow DIVIDE [28] to recover the missing views. We conduct experiments on four widely-used incomplete MvC benchmarks and compare CANDY with other baseline methods [51, 52, 55–57, 18, 58–60, 28]. As demonstrated in Table 3, CANDY could achieve competitive results comparable to SOTA methods, even though it is primarily designed for handing DNC rather than missing modalities.

### 4.3 Ablation Studies and Parameter Analysis (Component Indispensability)

In this section, we conduct ablation studies and parameter analysis to investigate the indispensable role and robustness of our modules.

As shown in Table 4, we design the following four method variants for the ablation studies: i) *Warmup Only*: using the identity matrix $\mathbf{I}$ as the target for Eq. 1 throughout the training process; ii) *Re-alignment*: adopting re-alignment strategy like the PVP studies; iii) *SCD*: performing the SCD module to denoise the vanilla affinity graph $\hat{\mathbf{A}}^{(v_1 \to v_2)}$ and using the resulting graph as the target for Eq. 1. iv) *CSM*: the complete version of CANDY, adopting the CSM module to induce $\mathbf{G}^{(v_1 \to v_2)}$ for recalling the false negatives and performing Eq. 6 to obtain the final pseudo target. From the results, one could observe that both the SCD and CSM modules play important roles in achieving robustness against DNC.

Table 2: Clustering performance comparisons on five widely-used multi-view datasets. The results are the mean of five individual runs. The best and second best results are shown in **bold** and underline, respectively.

| FP Ratio | Methods | Scene15 | | | Caltech-101 | | | LandUse21 | | | Reuters | | | NUS-WIDE | | | Average | | |
|---|---|---|---|---|---|---|---|---|---|---|---|---|---|---|---|---|---|---|---|
| | | ACC | NMI | ARI | ACC | NMI | ARI | ACC | NMI | ARI | ACC | NMI | ARI | ACC | NMI | ARI | ACC | NMI | ARI |
| 0% | DCCAE (ICML'15) | 34.6 | 39.0 | 19.7 | 45.8 | 68.6 | 37.7 | 15.6 | 24.4 | 4.4 | 42.0 | 20.3 | 8.5 | 47.5 | 17.1 | 37.6 | 37.1 | 33.9 | 21.6 |
| | BMVC (TPAMI'18) | 40.5 | 41.2 | 24.1 | 50.1 | 72.4 | 33.9 | 25.3 | 28.6 | 11.4 | 42.4 | 21.9 | 15.1 | 36.0 | 21.0 | 16.5 | 38.9 | 37.0 | 20.2 |
| | PVC (NeurIPS'20) | 38.0 | 39.8 | 21.1 | 20.5 | 51.4 | 15.7 | 16.8 | 25.2 | 5.6 | 44.1 | 27.1 | 27.1 | 19.3 | 7.7 | 3.8 | 27.7 | 30.2 | 14.7 |
| | MVCLN (CVPR'21) | 37.9 | 42.3 | 25.6 | 39.6 | 65.3 | 32.8 | 26.1 | 30.7 | 12.5 | 38.8 | **42.1** | 25.2 | 54.1 | 38.3 | 35.7 | 39.3 | 43.7 | 26.4 |
| | SURE (TPAMI'23) | 41.0 | 43.2 | 25.0 | 43.8 | 70.1 | 29.5 | 25.1 | 28.3 | 10.9 | 49.1 | 29.9 | 23.6 | 57.4 | 44.8 | 38.3 | 43.3 | 43.3 | 25.5 |
| | GCFAgg (CVPR'23) | 42.2 | 42.5 | 24.4 | 56.6 | 80.7 | 37.9 | 27.5 | 31.3 | 14.0 | 34.4 | 23.8 | 10.5 | 41.1 | 32.1 | 18.6 | 40.4 | 42.1 | 21.1 |
| | CGCN (TCSVT'24) | 42.9 | 43.4 | 25.0 | 49.1 | 75.2 | 33.8 | 28.8 | 36.0 | 15.0 | 45.8 | 27.0 | 22.3 | 61.2 | 48.1 | **41.2** | 45.6 | 45.9 | 27.5 |
| | DIVIDE (AAAI'24) | **49.1** | **48.7** | **31.6** | 62.2 | 83.0 | 50.5 | **32.3** | **39.7** | **18.1** | **59.3** | 39.5 | 29.0 | 45.1 | 30.9 | 19.4 | 49.6 | **48.4** | 29.7 |
| | CANDY (Ours) | 42.0 | 41.6 | 24.7 | **67.3** | **83.8** | **60.0** | 30.6 | 36.5 | 16.2 | 57.7 | 30.8 | **37.1** | **62.1** | **49.0** | 37.0 | **51.9** | 48.3 | **35.0** |
| 20% | DCCAE (ICML'15) | 32.9 | 17.1 | **29.6** | 36.9 | 39.2 | **60.1** | 15.0 | 3.8 | **17.4** | 41.6 | 13.1 | 19.3 | 41.6 | 11.6 | 26.9 | 33.6 | 17.0 | 30.7 |
| | BMVC (TPAMI'18) | 20.0 | 10.2 | 4.7 | 42.7 | 58.2 | 24.6 | 16.1 | 13.0 | 4.3 | 36.4 | 11.9 | 8.1 | 27.7 | 10.7 | 7.7 | 28.6 | 20.8 | 9.9 |
| | PVC (NeurIPS'20) | 31.2 | 25.5 | 13.6 | 8.3 | 30.2 | 3.8 | 22.8 | 28.0 | 8.4 | 32.4 | 15.4 | 15.3 | 34.3 | 22.2 | 13.6 | 25.8 | 24.3 | 10.9 |
| | MVCLN (CVPR'21) | 39.3 | 36.7 | 21.7 | 43.3 | 64.0 | 52.8 | 24.4 | 26.1 | 10.8 | 37.9 | 35.9 | 20.3 | 42.5 | 29.3 | 21.3 | 37.5 | 38.4 | 25.4 |
| | SURE (TPAMI'23) | 40.0 | 37.3 | 21.5 | 26.9 | 49.9 | 18.0 | 25.2 | 27.4 | 11.6 | 40.7 | 20.9 | 15.8 | 57.0 | 45.0 | **38.6** | 38.0 | 36.1 | 21.1 |
| | GCFAgg (CVPR'23) | 40.9 | 38.6 | 22.7 | 50.1 | 70.6 | 30.1 | 25.7 | 27.8 | 11.9 | 35.2 | 19.0 | 10.8 | 38.6 | 23.3 | 15.6 | 38.1 | 35.9 | 18.2 |
| | CGCN (TCSVT'24) | 40.7 | 38.0 | 22.1 | 40.8 | 64.9 | 27.2 | 27.0 | 31.4 | 13.3 | 43.5 | 23.0 | 19.4 | 58.0 | 41.7 | 35.9 | 42.0 | 39.8 | 23.6 |
| | DIVIDE (AAAI'24) | **42.4** | 39.9 | 24.5 | 48.3 | 69.1 | 38.0 | 30.9 | 35.1 | 16.2 | **55.3** | **36.9** | **31.0** | 44.9 | 28.3 | 18.2 | 44.4 | 41.9 | 25.6 |
| | CANDY (Ours) | 40.4 | **40.3** | 23.7 | **65.9** | **82.3** | 60.1 | 30.5 | **35.3** | 15.7 | 54.2 | 27.9 | 33.8 | **60.3** | **47.1** | 36.9 | **50.3** | **46.6** | **34.0** |
| 50% | DCCAE (ICML'15) | 26.8 | 10.2 | 19.8 | 27.0 | 26.8 | 49.8 | 13.3 | 2.8 | 13.2 | 37.7 | 9.2 | 12.5 | 32.3 | 7.1 | 13.5 | 27.4 | 11.2 | 21.8 |
| | BMVC (TPAMI'18) | 13.6 | 3.9 | 1.4 | 26.5 | 34.2 | 8.9 | 13.5 | 7.5 | 1.9 | 26.6 | 3.3 | 2.3 | 18.4 | 3.1 | 1.9 | 19.7 | 10.4 | 3.3 |
| | PVC (NeurIPS'20) | 20.3 | 10.2 | 13.6 | 7.4 | 21.8 | 5.0 | 20.6 | 28.5 | 8.7 | 42.9 | 23.5 | 23.4 | 24.1 | 10.1 | 9.9 | 23.1 | 18.8 | 12.1 |
| | MVCLN (CVPR'21) | **41.3** | 19.7 | 15.1 | 21.4 | 39.1 | 11.7 | 21.4 | 21.8 | 7.8 | 34.8 | **35.5** | 19.7 | 31.7 | 16.6 | 10.7 | 30.1 | 26.5 | 13.0 |
| | SURE (TPAMI'23) | 37.1 | 35.7 | 20.3 | 19.9 | 41.7 | 13.2 | 23.1 | 22.8 | 8.9 | 38.0 | 18.5 | 14.3 | 35.0 | 17.4 | 12.0 | 30.6 | 27.2 | 13.7 |
| | GCFAgg (CVPR'23) | 34.1 | 32.9 | 17.3 | 42.2 | 63.0 | 24.8 | 25.2 | 24.9 | 10.9 | 28.5 | 8.9 | 4.5 | 26.7 | 10.5 | 6.4 | 31.3 | 28.0 | 12.8 |
| | CGCN (TCSVT'24) | 32.5 | 29.5 | 15.7 | 33.4 | 59.3 | 21.6 | 25.8 | 28.2 | 11.9 | 40.5 | 16.1 | 14.1 | 50.1 | 33.8 | 27.4 | 36.5 | 33.4 | 18.1 |
| | DIVIDE (AAAI'24) | 37.4 | 34.0 | 20.3 | 39.1 | 58.7 | 32.5 | 28.1 | 30.4 | 13.5 | 41.2 | 19.4 | 14.8 | 44.0 | 23.9 | 16.6 | 38.0 | 33.3 | 19.5 |
| | CANDY (Ours) | **41.3** | **39.4** | **24.0** | **60.7** | **79.0** | **56.6** | **29.9** | **33.1** | **15.2** | **47.4** | 21.7 | **27.3** | **58.1** | **43.2** | **34.5** | **47.5** | **43.3** | **31.5** |
| 80% | DCCAE (ICML'15) | 20.9 | 6.7 | 14.4 | 18.4 | 15.8 | 41.8 | 14.5 | 3.2 | 13.4 | 35.3 | 7.6 | 10.0 | 36.2 | 14.9 | 21.9 | 25.1 | 9.6 | 20.3 |
| | BMVC (TPAMI'18) | 10.5 | 1.5 | 0.3 | 11.9 | 18.3 | 1.5 | 10.1 | 4.2 | 0.4 | 21.3 | 0.5 | 0.1 | 13.1 | 0.6 | 0.2 | 13.4 | 5.0 | 0.5 |
| | PVC (NeurIPS'20) | 20.3 | 10.2 | 4.6 | 7.5 | 20.8 | 4.2 | 22.5 | 29.3 | 9.3 | 35.7 | 13.2 | 13.1 | 19.3 | 7.7 | 3.8 | 21.1 | 16.2 | 7.0 |
| | MVCLN (CVPR'21) | 35.7 | 16.2 | 13.9 | 13.9 | 34.2 | 10.9 | 17.0 | 15.7 | 4.4 | 24.3 | **28.1** | 12.4 | 24.3 | 10.0 | 5.7 | 23.0 | 20.8 | 9.5 |
| | SURE (TPAMI'23) | 27.4 | 30.7 | 14.2 | 16.2 | 38.3 | 9.0 | 18.0 | 17.6 | 5.5 | 34.6 | 15.5 | 13.0 | 23.7 | 9.4 | 5.4 | 24.0 | 22.3 | 9.4 |
| | GCFAgg (CVPR'23) | 26.5 | 24.8 | 11.4 | 26.7 | 45.5 | 12.6 | 22.4 | 23.0 | 8.7 | 25.6 | 4.6 | 2.7 | 17.0 | 3.0 | 1.5 | 23.6 | 20.2 | 7.4 |
| | CGCN (TCSVT'24) | 28.7 | 24.0 | 12.5 | 21.3 | 46.6 | 13.2 | 25.2 | 27.7 | 11.4 | 29.0 | 7.9 | 6.5 | 50.1 | 34.6 | 28.0 | 30.9 | 28.2 | 14.3 |
| | DIVIDE (AAAI'24) | 34.4 | 30.4 | 18.3 | 27.8 | 50.8 | 21.1 | 27.1 | 28.1 | 12.8 | **41.1** | 24.7 | **19.5** | 45.8 | 28.3 | 19.1 | 35.2 | 32.5 | 18.2 |
| | CANDY (Ours) | **38.8** | **36.6** | **20.7** | **52.6** | **76.8** | **52.9** | **28.1** | **31.3** | **13.5** | 37.0 | 12.4 | 15.6 | **55.6** | **39.1** | **32.6** | **42.4** | **39.2** | **27.1** |

Table 3: Clustering performance on incomplete multi-view datasets, in which 50% of samples are with missing views. The results are the mean of five individual runs. The best and second best results are shown in **bold** and underline, respectively.

| Methods | Scene15 | | | Caltech101 | | | Reuters | | | LandUse21 | | | Average | | |
|---|---|---|---|---|---|---|---|---|---|---|---|---|---|---|---|
| | ACC | NMI | ARI | ACC | NMI | ARI | ACC | NMI | ARI | ACC | NMI | ARI | ACC | NMI | ARI |
| DCCAE (ICML'15) | 29.0 | 29.1 | 12.9 | 29.1 | 58.8 | 23.4 | 47.0 | 28.0 | 14.5 | 14.9 | 20.9 | 3.7 | 30.0 | 34.2 | 13.6 |
| BMVC (TPAMI'18) | 32.5 | 30.9 | 11.6 | 40.0 | 58.5 | 10.2 | 32.1 | 7.0 | 2.9 | 18.8 | 18.7 | 3.7 | 30.9 | 28.8 | 7.1 |
| PMVC (AAAI'14) | 25.5 | 25.4 | 11.3 | 50.3 | 74.5 | 41.5 | 29.3 | 7.4 | 4.4 | 20.0 | 23.6 | 8.0 | 31.3 | 32.7 | 16.3 |
| DAIMC (IJCAI'18) | 27.0 | 23.5 | 10.6 | 56.2 | 78.0 | 41.8 | 40.9 | 18.7 | 15.0 | 19.3 | 19.5 | 5.8 | 35.9 | 34.9 | 18.3 |
| EERIMVC (TPAMI'20) | 28.9 | 27.0 | 8.4 | 43.6 | 69.0 | 26.4 | 29.8 | 12.0 | 4.2 | 22.1 | 25.2 | 9.1 | 31.1 | 33.3 | 12.0 |
| SURE (TPAMI'22) | 39.6 | 41.6 | 23.5 | 34.6 | 57.8 | 19.9 | 47.2 | 30.9 | 23.3 | 23.1 | 28.6 | 10.6 | 36.1 | 39.7 | 19.3 |
| DSIMVC (ICML'22) | 30.6 | 35.5 | 17.2 | 16.4 | 24.8 | 9.2 | 39.9 | 19.6 | 17.1 | 18.6 | 18.8 | 5.7 | 26.4 | 24.7 | 12.3 |
| DCP (TPAMI'22) | 39.5 | 42.4 | 23.5 | 44.3 | 71.0 | 45.3 | 34.6 | 17.5 | 2.9 | 22.2 | 27.0 | 10.4 | 35.2 | 39.5 | 20.5 |
| ProImp (IJCAI'23) | 41.6 | 42.9 | 25.3 | 36.3 | 65.4 | 25.4 | 51.9 | 35.5 | 28.5 | 22.4 | 26.6 | 9.9 | 38.1 | 42.6 | 22.3 |
| DIVIDE (AAAI'24) | **46.8** | **45.7** | **29.1** | 63.4 | 82.5 | 52.4 | **54.7** | **37.3** | **28.6** | **30.0** | **35.8** | **16.0** | **48.7** | **50.3** | 31.5 |
| CANDY (Ours) | 40.0 | 40.2 | 24.1 | **69.5** | **83.9** | **65.5** | 54.2 | 34.8 | 27.2 | 28.8 | 31.1 | 14.4 | 48.1 | 47.5 | **32.8** |

Table 4: Ablation studies on the Caltech-101 and NUS-WIDE datasets with FP ratio of $20\%$ and $50\%$. ✓ represents using this component.

| FP Ratio | Warmup Only | Re-alignment | SCD | CSM | Caltech-101 | | | NUS-WIDE | | |
|---|---|---|---|---|---|---|---|---|---|---|
| | | | | | ACC | NMI | ARI | ACC | NMI | ARI |
| 20% | ✓ | | | | 46.9 | 67.5 | 29.5 | 57.5 | 37.9 | 33.1 |
| | ✓ | ✓ | | | 49.9 | 70.9 | 32.3 | 58.6 | 39.8 | 34.7 |
| | ✓ | ✓ | ✓ | | 56.5 | 78.4 | 38.4 | 58.1 | 43.6 | 37.0 |
| | ✓ | ✓ | ✓ | ✓ | 65.0 | 82.3 | 60.1 | 60.3 | 47.1 | 36.9 |
| 50% | ✓ | | | | 35.6 | 54.2 | 22.5 | 44.2 | 21.0 | 17.2 |
| | ✓ | ✓ | | | 41.1 | 60.3 | 26.3 | 46.6 | 23.8 | 19.8 |
| | ✓ | ✓ | ✓ | | 54.0 | 76.6 | 36.2 | 55.6 | 40.9 | 35.0 |
| | ✓ | ✓ | ✓ | ✓ | 60.7 | 79.0 | 56.6 | 58.1 | 43.2 | 34.5 |

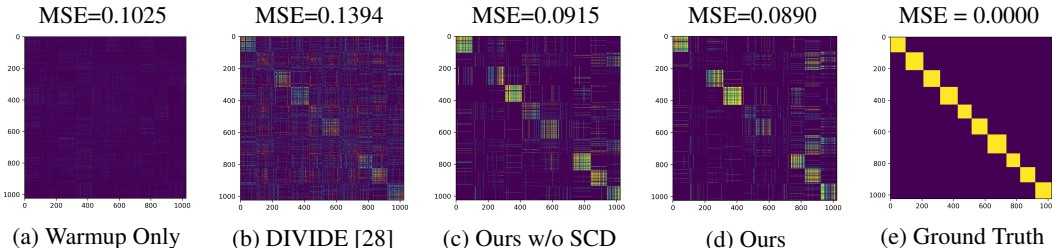

(a) Warmup only     (b) Ours (50 epochs)     (c) Ours (Converged)

Figure 3: The normalized similarity distribution of true positive and false positive pairs.

Figures 5a and 5b demonstrate that our method is robust to the selection of the denoising hyper-parameter $\eta$. Notably, setting $\eta$ too high would destroy the structural information of the high-order graph $\mathbf{G}^{(v_1 \rightarrow v_2)}$. Therefore, we fix $\eta$ at $0.2$ for all experiments without elaborated tuning.

### 4.4 Visualization on the Robustness (Working Mechanism)

To shed light on the working mechanism behind CANDY, we visualize the achieved robustness against the false positive and negative correspondences, respectively. Fig. 3 depicts the distribution of true and false positive pairs, where one could observe that the SCD module could remarkably distinguish the noisy correspondence from the clean one, thus supporting the robustness against false positive correspondence. Meanwhile, Fig. 4 presents the false negative recalling effects of different method variants, which demonstrate the significant semantic mining capacity of our CSM module and the polishing ability of the SCD module.

MSE=0.1025    MSE=0.1394    MSE=0.0915    MSE=0.0890    MSE = 0.0000

(a) Warmup Only   (b) DIVIDE [28]   (c) Ours w/o SCD   (d) Ours   (e) Ground Truth

Figure 4: The visualization of the cross-view similarity matrix, where each block is ordered using the ground-truth labels. For quantitative comparisons, we report the MSE between each result and the ground truth.

### 4.5 Comparisons with Noisy Correspondence Learning Approach (Approach Necessity)

As claimed in Related Works, we argue that the existing noisy correspondence learning cannot address the DNC problem well. In this section, we verify the necessity to devise a new approach

Table 5: Performance comparisons between the SOTA noisy correspondence learning method (namely, RCL) and CANDY on handling the DNC problem. For a fair comparison, we adopt the same backbone (DIVIDE) for RCL as used in CANDY.

| dataset | **Caltech-101** | | | | | | **NUS-WIDE** | | | | | |
|---|---|---|---|---|---|---|---|---|---|---|---|---|
| method | DIVIDE[28]+RCL[61] | | | DIVIDE+Ours | | | DIVIDE+RCL | | | DIVIDE+Ours | | |
| FP ratio | ACC | NMI | ARI | ACC | NMI | ARI | ACC | NMI | ARI | ACC | NMI | ARI |
| 0.0 | 44.9 | 70.5 | 28.0 | **67.3** | **83.8** | **60.0** | 61.0 | 45.4 | 40.6 | **62.1** | **49.0** | **37.0** |
| 0.2 | 38.4 | 59.8 | 21.0 | **65.9** | **82.3** | **60.1** | 53.2 | 35.8 | 29.9 | **60.3** | **47.1** | **36.9** |
| 0.5 | 27.6 | 44.5 | 12.0 | **60.7** | **79.0** | **56.6** | 36.4 | 19.8 | 13.9 | **58.1** | **43.2** | **34.5** |
| 0.8 | 16.7 | 32.1 | 8.2 | **52.6** | **76.8** | **52.9** | 22.0 | 6.5 | 3.6 | **55.6** | **39.1** | **32.6** |

to the DNC problem. To this end, we adopt the SOTA noisy correspondence learning method [61] in the cross-modal retrieval area for the MvC task using the same architecture (namely, DIVIDE) as CANDY. Table 5 summarizes the comparison results, highlighting the necessity of developing DNC-robust methods for contrastive MvC.

### 4.6 Study on the Generalizability (Approach Generalizability)

CANDY aims at generating a DNC-robust pseudo target for the existing contrastive MvC methods. To verify the generalizability of CANDY, in this section, we apply CANDY on another contrastive MvC baseline, namely, AECoKM. The results of "AECoKM" and "AECoKM+Ours" are shown in Fig. 5c, where the two methods are conducted with the false positive ratio varying from $0.0$ to $0.9$ with an interval of $0.1$. As one can observe, our CANDY could remarkably enhance the robustness and effectiveness of the baseline, demonstrating the plug-and-play role of our method.

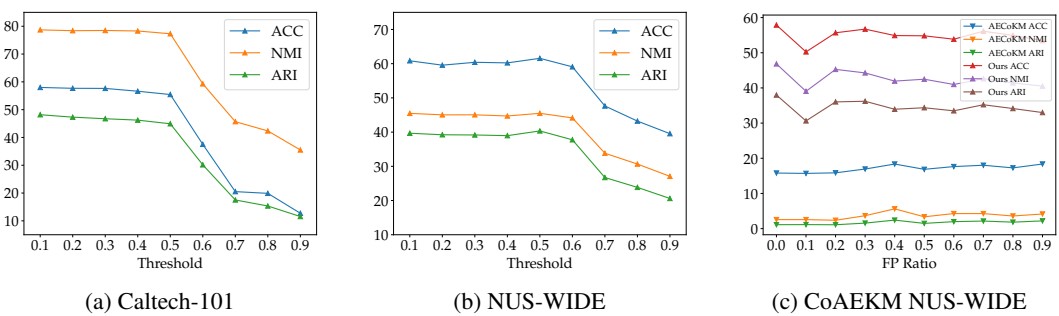

(a) Caltech-101    (b) NUS-WIDE    (c) CoAEKM NUS-WIDE

Figure 5: (a-b) Sensitivity studies of CANDY on the hyper-parameter $\eta$ for spectral denoising. (c) Investigation of the plug-and-play role and robustness of CANDY, where AECoKM is another contrastive multi-view clustering (MvC) baseline to which we transferred CANDY.

## 5   Conclusion

In this paper, we reveal and study a novel and practical problem within the field of contrastive Multi-view Clustering (MvC): Dual Noisy Correspondence (DNC). In brief, DNC involves both the false positive correspondences that arise during data collection, and the false negative correspondences that are inherent in the random sampling of contrastive MvC. To address this issue, we present CANDY comprising two novel modules: Context-based Semantic Mining (CSM) and Spectral-based Correspondence Denoising (SCD). On the one hand, CSM dexterously leverages contextual information to transform distinct views into a common contextual affinity space, thereby uncovering the semantically-associated false negatives. On the other hand, SCD refines the pseudo target to mitigate the adverse impact of false positives by using the spectral denoising technique. By integrating these models, our method provides a plug-and-play solution that could enhance the robustness of the most contrastive MvC methods against DNC. Extensive experiments on a broad spectrum of scenarios have validated the effectiveness of CANDY. In the future, we plan to extend CANDY to address more practical scenarios, such as simultaneously handling both noisy correspondence and missing modalities.

## Acknowledgments

This work was supported in part by NSFC under Grant 62176171, U21B2040, 623B2075, and 62472295; in part by the Fundamental Research Funds for the Central Universities under Grant CJ202303 and CJ202403; and in part by Sichuan Science and Technology Planning Project under Grant 24NSFTD0130 and 2024NSFTD0047.

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
