# OpenReview forum: "Robust Contrastive Multi-view Clustering against Dual Noisy Correspondence"
_NeurIPS.cc/2024/Conference — NeurIPS 2024 poster_

### Official Review · Reviewer_G6Ay · 2024-07-03

**Soundness:** 3
**Presentation:** 3
**Contribution:** 3
**Rating:** 7
**Confidence:** 5

**Summary:**

The paper presents a novel method, Contextually-spectral based correspondence refinery (CANDY), to address the Dual Noisy Correspondence (DNC) problem in contrastive multi-view clustering (MvC). CANDY utilizes inter-view similarities as context and employs a spectral-based module for denoising correspondence, effectively mitigating the influence of false positives and uncovering false negatives. The method's effectiveness is demonstrated through extensive experiments on five multi-view benchmarks, outperforming eight competitive MvC methods.

**Strengths:**

- CANDY introduces a new perspective to handle DNC, which is practical and challenging in MvC. This paper demonstrated potential capacity as a plug-and-play solution to enhance the robustness of existing contrastive MvC methods.
- The combination of context-based semantic mining and spectral-based denoising provides a robust solution against noisy correspondence. Extensive experiments and comparisons with state-of-the-art methods validate the effectiveness of CANDY.

**Weaknesses:**

1. How will the choice of the Gaussian kernel parameter $\sigma$ influence the performance of CANDY? The sensitivity of the clustering results to this parameter needs to be thoroughly examined, as different values of $\sigma $ might affect the affinity graph’s construction.
2. Can the authors provide insights into the computational overhead introduced by the CANDY method?
3. How are the false negative and false positive proportions handled in the experiments? Further clarification is needed on how this proportion is maintained across different datasets and experimental conditions. Does the false negative proportion keep fixed as 1/K in Table 1?
4. An interesting question arises from the paper’s claim that CANDY can serve as a plug-and-play solution. Is it possible to integrate the pseudo target generated by CANDY into other contrastive multi-view clustering methods beyond DIVIDE? Exploring this potential would demonstrate the versatility of CANDY and its applicability to a broader range of clustering techniques, providing valuable insights into how the pseudo target can enhance other methods’ robustness against noisy correspondences.

**Questions:**

Please refer to weaknesses.

**Limitations:**

Please refer to weaknesses.

---

> ### Author Rebuttal · Authors · 2024-08-07
>
> Thank you for your acknowledgment of our method. We will address your questions one by one.
>
> > ***Question 1**: **How will the choice of the Gaussian kernel parameter σ influence the performance** of CANDY? The sensitivity of the clustering results to this parameter needs to be thoroughly examined, as different values of σ might affect the affinity graph's construction.*
>
> Thank you for your suggestions. As you suggested, we conduct parameter analysis experiments to investigate the influence of $\sigma$ on the Caltech-101 dataset by varying the value of $\sigma$ from [0.01, 0.23] with an interval of 0.02. The results are summarized in the following table.
>
> | Dataset  \\ $\sigma$ | 0.01  | 0.03  | 0.05      | 0.07  | 0.09  | 0.11  | 0.13      | 0.15  | 0.17  | 0.19  | 0.21  | 0.23  |
> | -------------------- | ----- | ----- | --------- | ----- | ----- | ----- | --------- | ----- | ----- | ----- | ----- | ----- |
> | Caltech-101          | 16.13 | 16.75 | **60.97** | 60.71 | 60.31 | 51.41 | 47.22     | 41.97 | 37.14 | 34.72 | 32.37 | 32.98 |
>
> From the results, one could observe that our method performs stably in the range of [0.05, 0.09] for $\sigma$. Thus, we simply set the Gaussian kernel parameter $\sigma$ as 0.07 for all experiments in the paper. For a clear observation, we include the plot of the result as Figure S1 in the rebuttal PDF.
>
> > ***Question 2**: Can the authors provide insights into the **computational overhead** introduced by the CANDY method?*
>
> As you suggested, we run DIVIDE and our CANDY with the same network architecture for a fair comparison, and report their running time (in seconds, per epoch) in the following table.
>
> | Model        | Caltech101 | LandUse21 | NUSWIDE | Reuters | Scene15 |
> | ------------ | ---------- | --------- | ------- | ------- | ------- |
> | DIVIDE       | 0.1438     | 0.0647    | 0.2286  | 0.4617  | 0.1346  |
> | CANDY (Ours) | 2.4971     | 0.7010    | 2.1663  | 4.6905  | 1.2743  |
>
> As indicated in the table, CANDY takes approximately 10 times the computational time per epoch compared to DIVIDE. The additional computational overhead is mainly attributed to the Singular Value Decomposition (SVD) procedure in our approach. In the future, we plan to explore lightweight and efficient alternatives for SVD.
>
> > ***Question 3:** How are the false negative and false positive proportions handled in the experiments? Further clarification is needed on **how this proportion is maintained across different datasets and experimental conditions**. Does the false negative proportion keep fixed as 1/K in Table 1?*
>
> As per your suggestions, we have given the factual noise ratios for both false positives (FPs) and false negatives (FNs) in the following table. Specifically, we set one view as the anchor view, and simulate FP samples by randomly shuffling the other view by a given percentage, denoted by FP ratio $\eta$. FN is inherent to the dataset. For example, the FN ratio of a dataset is $1 / K$ if its samples are evenly distributed in $K$ classes.
>
> | $\eta$ | Caltech101 |      | LandUse21 |      | NUSWIDE |      | Reuters |       | Scene15 |      |
> | ------ | ---------- | ---- | --------- | ---- | ------- | ---- | ------- | ----- | ------- | ---- |
> |        | FP         | FN   | FP        | FN   | FP      | FN   | FP      | FN    | FP      | FN   |
> | 0.0    | 0.00       | 2.84 | 0.00      | 4.73 | 0.00    | 9.99 | 0.00    | 21.40 | 0.00    | 6.91 |
> | 0.2    | 19.34      | 2.84 | 19.10     | 4.73 | 17.98   | 9.99 | 15.67   | 21.40 | 18.68   | 6.91 |
> | 0.5    | 48.45      | 2.84 | 47.33     | 4.73 | 45.07   | 9.99 | 39.64   | 21.40 | 46.42   | 6.91 |
> | 0.8    | 77.48      | 2.84 | 76.24     | 4.73 | 72.03   | 9.99 | 62.57   | 21.40 | 73.98   | 6.91 |
>
> In the next version, we will detail the simulation procedure of noisy correspondence and the practical noise ratios in our experiments.
>
> > ***Question 4**: An interesting question arises from the paper's claim that CANDY can serve as a plug-and-play solution. Is it possible to **integrate the pseudo target generated by CANDY into other contrastive multi-view clustering methods** beyond DIVIDE?*
>
> Our research has considered the broader applicability of CANDY, and we have conducted experiments to demonstrate its integration with other methods beyond DIVIDE. In Section 4.6 of our manuscript, we present the results of integrating CANDY with another model, AECoKM, on the NUS-WIDE dataset. Figure 5(c) summarizes the performance improvements observed with AECoKM under different false positive (FP) ratios. For your convenience, we have converted the figure as a table and attached it below.
>
> | FP Ratio      |     | 0.0   | 0.1   | 0.2   | 0.3   | 0.4   | 0.5   | 0.6   | 0.7   | 0.8   | 0.9   |
> | ------------- | --- | ----- | ----- | ----- | ----- | ----- | ----- | ----- | ----- | ----- | ----- |
> | AECoKM        | ACC | 15.80 | 15.70 | 15.86 | 16.94 | 18.36 | 16.86 | 17.64 | 18.02 | 17.27 | 18.37 |
> |               | NMI | 2.56  | 2.56  | 2.38  | 3.67  | 5.62  | 3.36  | 4.28  | 4.27  | 3.59  | 4.16  |
> |               | ARI | 1.12  | 1.14  | 1.10  | 1.57  | 2.42  | 1.49  | 1.98  | 2.15  | 1.84  | 2.20  |
> | AECoKM + Ours | ACC | 57.88 | 50.25 | 55.68 | 56.72 | 54.89 | 54.82 | 53.85 | 56.12 | 54.93 | 53.44 |
> |               | NMI | 46.83 | 39.04 | 45.27 | 44.27 | 41.94 | 42.48 | 40.98 | 42.61 | 41.43 | 40.51 |
> |               | ARI | 37.98 | 30.63 | 36.03 | 36.21 | 33.94 | 34.37 | 33.49 | 35.22 | 34.14 | 33.00 |

---

> > ### Comment · Reviewer_G6Ay · 2024-08-09
> >
> > Thanks very much for the detailed response. The authors have provided solid and convincing experimental results to answer my problems.

---

> > > ### Author Response · Authors · 2024-08-12
> > >
> > > We sincerely appreciate your positive recognition and assessment of our work!

---

### Official Review · Reviewer_6oUm · 2024-07-03

**Soundness:** 3
**Presentation:** 3
**Contribution:** 3
**Rating:** 7
**Confidence:** 5

**Summary:**

The authors delve into the study of contrastive multi-view clustering (MVC) and aim to address the false positive and false negative correspondence issues, collectively referred to as dual noisy correspondence. To tackle this problem, they propose a two-fold solution named CANDY. Firstly, CANDY exploits inter-view similarities as context to uncover false negatives. Secondly, it employs a spectral-based module to denoise correspondences, thereby mitigating the negative impact of false positives.

One of the most interesting insights in this paper is the observation that context can serve as a new axis for transforming data similarity into a high-order affinity space.

To verify the effectiveness of CANDY, the authors conduct experiments across various datasets and settings.

**Strengths:**

i) This paper is well-written and structured, making it accessible to readers even outside the immediate research community. Additionally, the experiment designs are intriguing, and the extensive experiments convincingly demonstrate the effectiveness of the proposed methods and the two corresponding modules.

ii) The proposed method is technically sound. Moreover, it can serve as a plug-and-play module that can be integrated into other contrastive MVC methods, enhancing their robustness.

**Weaknesses:**

i) I have some doubts regarding the results in Figure 4. Firstly, how is the cross-view similarity matrix arranged? The current form appears to be arranged by the ground truth, whereas my understanding is that the ground truth should be agnostic during training. Secondly, how is robustness compared in the figure? While the proposed method can recall more false negative samples, it seems to introduce more wrongly recalled samples. How is the trade-off between truly and falsely recalled samples balanced?

ii)  Regarding the references in Line 31, is the observation in Fig. 1(a) first proposed in this paper?

iii) Two open-world questions. Firstly, robustness against incomplete views is another important topic in the MVC community. Some baselines in the paper address this issue. Can the proposed method be extended to handle incomplete views? If so, this would significantly enhance the paper’s contribution. Secondly, the approach is mainly presented in scenarios involving two views. How could the method be extended to handle more views (≥ 3)?

**Questions:**

Please see the weaknesses.

**Limitations:**

No.

---

> ### Author Rebuttal · Authors · 2024-08-07
>
> Thanks for your constructive reviews and suggestions. Below, we will address each of your questions.
>
> > ***Question 1.1**: I have some doubts regarding the results in Figure 4. Firstly, **how is the cross-view similarity matrix arranged**? The current form appears to be arranged by the ground truth, whereas my understanding is that the ground truth should be agnostic during training.*
>
> The cross-view similarity matrix depicted in Figure 4 is arranged according to the ground-truth labels. To be specific, we first compute the similarities of cross-view representations and then arrange them according to the labels. In other words, each diagonal block in the matrix represents the similarities of within-class samples. Thanks to the matrix re-arrangement manner, it allows us to investigate the false negatives recall capacity by comparing the consistency between the cross-view similarity matrix and the ground-truth one. Notably, the labels are only utilized for the above analysis experiment, while keeping agnostic during both the training and inference phases.
>
> > ***Question 1.2**: Secondly, **how is robustness compared in the figure**? While the proposed method can recall more false negative samples, it seems to introduce more wrongly recalled samples. How is the trade-off between truly and falsely recalled samples balanced?*
>
> To quantitatively compare the robustness between the baselines and our method, we additionally report the Mean Squared Error (MSE) to measure the difference between each cross-view similarity matrix and the ground-truth one in the figure. More specifically, the smaller MSE value indicates the greater robustness of the method against false negatives. Notably, as truly and falsely recalled samples in local regions cannot reflect the global difference, it is sub-optimal to measure the robustness by directly observing the cross-view similarity matrix. In contrast, the MSE value seeks a trade-off between truly and falsely recalled samples from the global perspective, thus being a promising metric to demonstrate the robustness against false negatives.
>
> > ***Question 2**: Regarding the references in Line 31, **is the observation in Fig. 1(a) first proposed in this paper**?*
>
> Yes, our work could be the first study to reveal the dual noisy correspondence (DNC) challenge in the multi-view clustering community. In brief, DNC refers to the noise present in both cross-view positive and negative pairs. Although the references in Line 31 reveal the noise in the training multimodal data, they either do not explicitly handle the noise [A,B,D,E], or only conquer false negative issue [C].
>
> [A] Learning transferable visual models from natural language supervision. International Conference on Machine Learning, 2021.
>
> [B] Scaling up visual and vision-language representation learning with noisy text supervision. International Conference on Machine Learning, 2021.
>
> [C] Robust multi-view clustering with incomplete information. IEEE Transactions on Pattern Analysis and Machine Intelligence, 2022.
>
> [D] Conceptual captions: A cleaned, hypernymed, image alt-text dataset for automatic image captioning. Proceedings of ACL, 2018.
>
> [E] HowTo100M: Learning a Text-Video Embedding by Watching Hundred Million Narrated Video Clips. International Conference on Computer Vision, 2019.
>
> > ***Question 3.1**: Firstly, robustness against incomplete views is another important topic in the MVC community. Some baselines in the paper address this issue. Can the proposed method be extended to **handle incomplete views**? If so, this would significantly enhance the paper's contribution.*
>
> Thank you for your constructive suggestions. As you suggested, we follow DIVIDE [F] to endow our method with the capacity of handling incomplete views. In brief, we utilize the cross-view decoders to recover the missing samples and perform k-means on all the latent representations to achieve incomplete multi-view clustering. The results are summarized in the following table, where the results of baseline methods are copied from the DIVIDE paper.
>
> | Incomplete (50% missing) | Model| Scene15||| Caltech101 ||| Reuters||| LandUse21 ||| Average|||
> | -- | -- | -- | -- | -- | -- | -- | -- | -- | -- | -- | -- | -- | -- | -- | -- | -- |
> ||| ACC| NMI| ARI| ACC| NMI| ARI| ACC| NMI| ARI| ACC | NMI| ARI| ACC| NMI| ARI|
> || SURE | 39.6 | 41.6 | 23.5 | 34.6 | 57.8 | 19.9 | 47.2 | 30.9 | 23.3 | 23.1| 28.6 | 10.6 | 36.1 | 39.7 | 19.3 |
> | | DIVIDE | **46.8** | **45.7** | **29.1** | 63.4 | 82.5 | 52.4 | **54.7** | **37.3** | **28.6** | **30.0**| **35.8** | **16.0** | **48.7** | **50.3** | 31.5 |
> || CANDY (Ours) | 40.0 | 40.2 | 24.1 | **69.5** | **83.9** | **65.5** | 54.2 | 34.8 | 27.2 | 28.8| 31.1 | 14.4 | 48.1 | 47.5 | **32.8** |
>
> From the results, one could observe that our method achieves promising performance in handling the incomplete view problem, even though it is primarily designed to address the dual noisy correspondence challenge.
>
> [F] Decoupled Contrastive Multi-view Clustering with High-order Random Walks. AAAI Conference on Artificial Intelligence, 2024.
>
> > ***Question 3.2**: Secondly, the approach is mainly presented in scenarios involving two views. **How could the method be extended to handle more views (≥ 3)**?*
>
> According to your comment, we extend CANDY to scenarios involving more than two views. Specifically, following [G], we conduct experiments on the Caltech7 dataset using two views (Caltech7-2V) and three views (Caltech7-3V). The results, summarized in the following table, demonstrate CANDY's capability in handling multiple views.
>
> | Method       | FP ratio | Caltech7-2V | Caltech7-3V |
> | ------------ | -------- | ----------- | ----------- |
> | CANDY (Ours) | 0.0      | 52.16       | 60.37       |
> |              | 0.5      | 41.96       | 47.93       |
>
> [G] Multi-level Feature Learning for Contrastive Multi-view Clustering. The IEEE / CVF Computer Vision and Pattern Recognition Conference, 2022.

---

> > ### Comment · Reviewer_6oUm · 2024-08-12
> >
> > Thank you for the author's response, which has effectively addressed my concerns. I will raise my score on the paper.

---

> > > ### Author Response · Authors · 2024-08-12
> > >
> > > Thank you for raising your score! We appreciate the time and effort you dedicated to reviewing this work.

---

### Official Review · Reviewer_L9C1 · 2024-07-06

**Soundness:** 3
**Presentation:** 3
**Contribution:** 3
**Rating:** 6
**Confidence:** 4

**Summary:**

This paper addresses a new problem called dual noisy correspondence, which the authors claim is practical and underexplored in the multi-view learning community. Dual noisy correspondence refers to two challenges: 1) false positive correspondences induced by irrelevant multi-view data and 2) false negative correspondences caused by the random sampling characteristic of contrastive learning. To address this problem, the authors propose a novel metric for cross-view similarity estimation that recalls more false negative pairs, and a spectral-based denoising method to address false positive correspondences. Extensive experiments on multiple datasets validate the superiority of the proposed method compared to existing approaches. Overall, this paper makes an important contribution by tackling the practical and underexplored problem of dual noisy correspondence in multi-view learning. The technical solutions proposed, including the new similarity metric and spectral denoising, demonstrate strong empirical performance.

**Strengths:**

The paper introduces a novel perspective on cross-view similarity measurement. Specifically, the authors observe that the row-wise similarity in the first-order cross-view similarity matrix, referred to as context in the paper, can serve as an effective metric for similarity estimation. Experiments show that this approach recalls more potential false negative pairs.
Extensive experiments on five benchmarks with four types of noise ratios comprehensively verify the effectiveness of the proposed method.

**Weaknesses:**

The authors claim to address a new problem, dual noisy correspondence (false positive and false negative). However, reviewers noted that similar issues have been explored in works [A] of other fields. Additional claims and discussions are needed to clarify the novelty of the proposed setting, which the authors regard as a significant contribution. [A] Cross-modal retrieval with noisy correspondence via consistency refining and mining, TIP.
-How about the time cost of the proposed method? It is essential to compare this with other methods, particularly the peer method (e.g., DIVIDE) with the same network architecture.
The reviewer suggests that the noise ratios for both false positive and false negative correspondences should be explicitly presented in the main table for a more comprehensive understanding of performance and comparisons.

**Questions:**

(1) What is the novelty of the proposed setting?

 (2) What is the additional time cost?

**Limitations:**

see weakness

---

> ### Author Rebuttal · Authors · 2024-08-07
>
> Thank you for your valuable review. We will address your questions one by one.
>
> > _**Question 1**: The authors claim to address a new problem, dual noisy correspondence (false positive and false negative). However, reviewers noted that similar issues have been explored in works [A] of other fields. Additional claims and discussions are needed to **clarify the novelty of the proposed setting**, which the authors regard as a significant contribution._
>
> Thank you for pointing out the related work [A], which addresses both the false positive and false negative correspondence for the image-text retrieval task. Although [A] shares some similarities with our work, the definitions of false positives and false negatives are remarkably different. First, [A] is specialized for the retrieval task where the false positives emerge i.f.f. the given two samples do not belong to the same instance. In contrast, in our work, false positives are defined as wrongly matched pairs consisting of samples from different classes. Second, [A] seeks to mine semantic-consistent samples and regards them as false negatives, whereas our work considers arbitrary cross-view within-class samples as false negatives. Intuitively, given a false positive/negative pair, our work has a probability of $\frac{1}{K}$ for both false positive re-alignment and false negative recalling, while the probabilities for [A] are $\frac{1}{N}$, where $K$ is the number of classes and $N$ is the number of instances in the dataset. Thanks to the differences, it is difficult or even impossible to realign mismatched pairs in [A], while our method can achieve this for a much smaller $K$.
>
> [A] Cross-modal Retrieval with Noisy Correspondence via Consistency Refining and Mining. IEEE Transactions on Image Processing, 2024.
>
> > ***Question 2**: How about the **time cost** of the proposed method? It is essential to compare this with other methods, particularly the peer method (e.g., DIVIDE) with the same network architecture.*
>
> As you suggested, we run DIVIDE and our CANDY with the same network architecture for a fair comparison, and report their running time (in seconds, per epoch) in the following table.
>
> | Model        | Caltech101 | LandUse21 | NUSWIDE | Reuters | Scene15 |
> | ------------ | ---------- | --------- | ------- | ------- | ------- |
> | DIVIDE       | 0.1438     | 0.0647    | 0.2286  | 0.4617  | 0.1346  |
> | CANDY (Ours) | 2.4971     | 0.7010    | 2.1663  | 4.6905  | 1.2743  |
>
> As indicated in the table, CANDY takes approximately 10 times the computational time per epoch compared to DIVIDE. The additional computational overhead is mainly attributed to the Singular Value Decomposition (SVD) procedure in our approach. In the future, we plan to explore lightweight and efficient alternatives for SVD.
>
> > ***Question 3**: The reviewer suggests that the **noise ratios for both false positive and false negative correspondences should be explicitly presented in the main table** for a more comprehensive understanding of performance and comparisons.*
>
> As per your suggestions, we have given the factual noise ratios for both false positives (FPs) and false negatives (FNs) in the following table. Specifically, we set one view as the anchor view, and simulate FP samples by randomly shuffling the other view by a given percentage, denoted by FP ratio $\eta$. FN is inherent to the dataset. For example, the FN ratio of a dataset is $1 / K$ if its samples are evenly distributed in $K$ classes.
>
> | $\eta$ | Caltech101 |      | LandUse21 |      | NUSWIDE |      | Reuters |       | Scene15 |      |
> | ------ | ---------- | ---- | --------- | ---- | ------- | ---- | ------- | ----- | ------- | ---- |
> |        | FP         | FN   | FP        | FN   | FP      | FN   | FP      | FN    | FP      | FN   |
> | 0.0    | 0.00       | 2.84 | 0.00      | 4.73 | 0.00    | 9.99 | 0.00    | 21.40 | 0.00    | 6.91 |
> | 0.2    | 19.34      | 2.84 | 19.10     | 4.73 | 17.98   | 9.99 | 15.67   | 21.40 | 18.68   | 6.91 |
> | 0.5    | 48.45      | 2.84 | 47.33     | 4.73 | 45.07   | 9.99 | 39.64   | 21.40 | 46.42   | 6.91 |
> | 0.8    | 77.48      | 2.84 | 76.24     | 4.73 | 72.03   | 9.99 | 62.57   | 21.40 | 73.98   | 6.91 |
>
> In the next version, we will detail the simulation procedure of noisy correspondence and the practical noise ratios in our experiments.

---

### Official Review · Reviewer_jTED · 2024-07-07

**Soundness:** 3
**Presentation:** 4
**Contribution:** 3
**Rating:** 6
**Confidence:** 4

**Summary:**

The manuscript "Robust Contrastive Multi-view Clustering against Dual Noisy Correspondence" addresses the Dual Noisy Correspondence (DNC) issue in contrastive multi-view clustering (MvC), where noise affects both positive and negative data pairs. The authors propose CANDY (Contextually-spectral based correspondence refinery), a method that uses inter-view similarities to identify false negatives and a spectral-based module to denoise correspondences, thus mitigating the impact of false positives. Extensive experiments on five multi-view benchmarks demonstrate CANDY's effectiveness over existing methods.

**Strengths:**

-The paper introduces the DNC problem, a novel and practical challenge in MvC, highlighting the dual nature of noise affecting both positive and negative pairs.

-CANDY combines two innovative components: the Context-based Semantic Mining (CSM) module, which identifies false negatives using high-order contextual affinities, and the Spectral-based Correspondence Denoising (SCD) module, inspired by signal processing techniques, to filter out false positives. This dual approach ensures robust clustering by comprehensively addressing both types of noise.

-The proposed method consistently outperforms state-of-the-art multi-view clustering methods on various benchmarks, demonstrating its robustness and effectiveness. The extensive experiments, varying the ratio of false positives, provide thorough validation of CANDY’s capabilities.

**Weaknesses:**

-The method's applicability to other types of multi-view learning tasks, such as classification or retrieval, is not explored, limiting its broader impact within the multi-view learning domain. Including classification results would enhance the paper.

-Visualization results, such as those in Figure 3, should be compared with other baseline methods to provide detailed insights into how CANDY's clustering performance or robustness visually differs from that of other methods, particularly in handling noisy correspondences.

**Questions:**

How is the pseudo target C defined in Equation 1 of the proposed method? Please provide a detailed explanation of its construction and role within the CANDY framework.

**Limitations:**

Please see weaknesses.

---

> ### Author Rebuttal · Authors · 2024-08-07
>
> Thanks for your constructive reviews and suggestions. We will address your questions one by one.
>
> > ***Question 1**: The method's **applicability to other types of multi-view learning tasks**, such as classification or retrieval, is not explored, limiting its broader impact within the multi-view learning domain. Including classification results would enhance the paper.*
>
> Thank you for your constructive suggestion. We follow [A] to endow our method with the capacity to handle the classification problem. To achieve multi-view classification, we randomly select 80% of the data as the training set, and use the remaining 20% as the test set. Then we simulate 50% noisy correspondence in the training set, while keeping the test set clean. After training the model using our CANDY method as the feature extractor, we train a support vector machine (SVM) on the features of the train set. Finally, we use the SVM to predict the labels on the features of the test set. The results are summarized in the following table.
>
> | Model | Caltech-101 | NUS-WIDE  |
> | - | - | - |
> | SURE [A] | 56.65 | 48.68 |
> | DIVIDE [B] | **76.12** | 50.06  |
> | CANDY (Ours) | 72.97 | **61.02** |
>
> Our method achieves competitive results on these datasets, justifying the applicability of our method to the classification task.
>
> [A] Robust Multi-View Clustering With Incomplete Information. IEEE Transactions on Pattern Analysis and Machine Intelligence, 2023.
>
> [B] Decoupled Contrastive Multi-view Clustering with High-order Random Walks. AAAI Conference on Artificial Intelligence, 2024.
>
> > ***Question 2**: **Visualization results, such as those in Figure 3, should be compared with other baseline methods** to provide detailed insights into how CANDY's clustering performance or robustness visually differs from that of other methods, particularly in handling noisy correspondences.*
>
> As per your suggestions, we have supplemented the visualization comparisons in Figures S2 and S3 of the rebuttal PDF file, comparing our CANDY with the most competitive baseline DIVIDE [B]. As observed from Figure S2, DIVIDE struggles to differentiate between true and false positives during training, indicating overfitting on noisy correspondence. In contrast, Figure S3 demonstrates that our CANDY achieves a clear separation between true and false positives, highlighting its superior robustness in handling noisy correspondences.
>
> > ***Question 3**: **How is the pseudo target C defined in Equation 1** of the proposed method? Please provide a detailed explanation of its construction and role within the CANDY framework.*
>
> We apologize for the unclear description regarding the construction of the pseudo target $\mathbf{C}$. $\mathbf{C}$ serves as the target of the contrastive loss, which is defined as
> $$
> \mathcal{L} = \sum_{v_1=1}^V \sum_{\substack{v_2=1 \\ v_2 \neq v_1}}^V \mathcal{H}\left(\mathbf{C}^{(v_1,v_2)}, \rho\left(\mathbf{Z}^{(v_1)}{\mathbf{Z}^{(v_2)}}^\top \right)\right),
> $$
> where $\mathcal{H}$ denotes the row-wise cross-entropy function with mean reduction, $\rho\left(\cdot \right)$ denotes the softmax function, $\mathbf{C}^{(v_1,v_2)} \in \mathbb{R}^{n \times n}$ and $\mathbf{Z}^{(v_1)}{\mathbf{Z}^{(v_2)}}^\top$ represent the pseudo targets and affinity matrix between views $v_1$ and $v_2$, respectively. In general, traditional contrastive multi-view clustering methods assume that the cross-view correspondence is faultless, typically adopting an identity matrix $\mathbf{I}\in \mathbb{R}^{n \times n}$ as the target. As verified in our experiments, such a vanilla target not only misleads the model to overfit false positives but also neglects numerous semantically associated false negatives. Therefore, we propose the following formulation of $\mathbf{C}$ to achieve robustness against such a dual noisy correspondence problem:
> $$
> \mathbf{C} = \widetilde{\mathbf{G}} + \varepsilon \mathbf{I},
> $$
> where $\varepsilon$ is a hyper-parameter set to 0.2 for all the experiments, and $\widetilde{\mathbf{G}}$ is the denoised pseudo target obtained through our context-based semantic mining and spectral-based correspondence denoising modules as elaborated in the manuscript. For your convenience, we attach the key steps for constructing the denoised target $\widetilde{\mathbf{G}}$ below. The pseudo target $\widetilde{\mathbf{G}}$ is obtained by
> $$
> \widetilde{\mathbf{G}}^{(v_1 \rightarrow v_2)} = \mathbf{U} \mathbf{\widetilde{\Sigma}} \mathbf{V^{\top}},
> $$
>
> where $\widetilde{\mathbf{\Sigma}} = \mathrm{diag}(\lambda_1, \ldots, \lambda_L, 0, \ldots, 0) \in \mathbb{R}^{n \times n}$ is a diagonal matrix consisting of the retained singular values $( \lambda_1 > \cdots > \lambda_L \ge \eta)$, with a denoising hyper-parameter $\eta$ set to $0.2$ in our experiments. To obtain $\widetilde{\mathbf{\Sigma}}$, $\mathbf{U}$ and $\mathbf{V}$, we decompose $\mathbf{G}^{(v_1 \rightarrow v_2)}$ by Singular Value Decomposition (SVD) as
>
> $$
> \mathbf{G}^{(v_1 \rightarrow v_2)} = \mathbf{U} \mathbf{\Sigma} \mathbf{V^{\top}},
> $$
> where $\mathbf{\Sigma} = \mathrm{diag}(\lambda_1, \ldots, \lambda_n)$ denotes a diagonal matrix consisting of the singular values, and columns of $\mathbf{U}$ and $\mathbf{V}$ are the left- and right-singular vectors, respectively. The cross-view high-order affinity graph $\mathbf{G}^{(v_1 \rightarrow v_2)}$ is defined as
> $$
> \mathbf{G}^{(v_1 \rightarrow v_2)} = \mathbf{A}^{(v_1 \rightarrow v_2)} {\mathbf{A}^{(v_2 \rightarrow v_2)}}^{\top},
> $$
>
> where the affinity graph $\mathbf{A}^{(v_1 \rightarrow v_2)}$ from $v_1$ to $v_2$ is constructed by samples $\mathbf{Z}^{(v_1)}$ and $\mathbf{Z}^{(v_2)}$ as nodes in a minibatch, with edge weights defined by Gaussian kernel similarity. Specifically, we formulate the connection probability from one node to all others as a context, which serves as an embedding of the node $i$ for semantic mining, with
> $$
> \mathbf{A}_{i j}^{(v_1 \rightarrow v_2)}=\exp (-\|[\mathbf{Z}^{(v_1)}]_i-[\mathbf{Z}^{(v_2)}]_j\|^2 / \sigma).
> $$

---

> > ### Comment · Reviewer_jTED · 2024-08-13
> >
> > The authors have addressed my concerns; therefore, I will maintain my score.

---

> > > ### Author Response · Authors · 2024-08-13
> > >
> > > Thank you for your recognition and positive assessment of our work. We sincerely appreciate your time and effort.

---

### Author Rebuttal · Authors · 2024-08-07

Dear ACs and Reviewers,

We sincerely appreciate your time and effort in reviewing our paper and providing constructive feedback. We thank the reviewers for your recognition of our novelty and contributions.

* This method serves as a plug-and-play module that can be integrated into other contrastive MVC methods [6oUm, G6Ay]
* The proposed method consistently outperforms state-of-the-art multi-view clustering methods on various benchmarks [jTED, L9C1, G6Ay].
* This paper is well-written and structured [6oUm].

In the following sections, we have addressed each concern and query raised by the reviewers during the rebuttal phase. We have also included a **PDF document with additional figures** to complement our responses.

Thank you again for your valuable feedback and time investment.

Best regards,

The Authors

---

### Decision · Program_Chairs · 2024-09-25

**Decision:**

Accept (poster)

**Comment:**

This paper makes a valuable contribution to the multi-view clustering (MvC) community by introducing the Dual Noisy Correspondence (DNC) problem and proposing a novel solution based on a contextually-spectral mechanism. The work is supported by strong experimental validation and demonstrates significant technical innovation. During the rebuttal process, the authors effectively addressed the reviewers' concerns, leading two out of four reviewers to raise their scores to clear acceptance. Overall, there is no dispute regarding the paper's contribution and technical novelty, and I recommend its acceptance.